# Therapeutic Treatment of Superoxide Dismutase 1 (G93A) Amyotrophic Lateral Sclerosis Model Mice with Medical Ozone Decelerates Trigeminal Motor Neuron Degeneration, Attenuates Microglial Proliferation, and Preserves Monocyte Levels in Mesenteric Lymph Nodes

**DOI:** 10.3390/ijms23063403

**Published:** 2022-03-21

**Authors:** Michael Bette, Eileen Cors, Carolin Kresse, Burkhard Schütz

**Affiliations:** 1Institute of Anatomy and Cell Biology, Philipps-University, 35037 Marburg, Germany; eileen.cors@gmail.com (E.C.); carolinkresse@gmx.de (C.K.); 2Department of Mitochondrial Proteostasis, Max-Planck-Institute for Biology of Ageing, 50931 Cologne, Germany

**Keywords:** amyotrophic, ozone, neurodegeneration, neuroinflammation, adaptive immune system, monocyte

## Abstract

Amyotrophic lateral sclerosis (ALS) is an incurable and lethal neurodegenerative disease in which progressive motor neuron loss and associated inflammation represent major pathology hallmarks. Both the prevention of neuronal loss and neuro-destructive inflammation are still unmet challenges. Medical ozone, an ozonized oxygen mixture (O_3_/O_2_), has been shown to elicit profound immunomodulatory effects in peripheral organs, and beneficial effects in the aging brain. We investigated, in a preclinical drug testing approach, the therapeutic potential of a five-day O_3_/O_2_
*i.p*. treatment regime at the beginning of the symptomatic disease phase in the superoxide dismutase (SOD1^G93A^) ALS mouse model. Clinical assessment of SOD1^G93A^ mice revealed no benefit of medical ozone treatment over sham with respect to gross body weight, motor performance, disease duration, or survival. In the brainstem of end stage SOD1^G93A^ mice, however, neurodegeneration was found decelerated, and SOD1-related vacuolization was reduced in the motor trigeminal nucleus in the O_3_/O_2_ treatment group when compared to sham-treated mice. In addition, microglia proliferation was less pronounced in the brainstem, while the hypertrophy of astroglia remained largely unaffected. Finally, monocyte numbers were reduced in the blood, spleen, and mesenteric lymph nodes at postnatal day 60 in SOD1^G93A^ mice. A further decrease in monocyte numbers seen in mesenteric lymph nodes from sham-treated SOD1^G93A^ mice at an advanced disease stage, however, was prevented by medical ozone treatment. Collectively, our study revealed a select neuroprotective and possibly anti-inflammatory capacity for medical ozone when applied as a therapeutic agent in SOD1^G93A^ ALS mice.

## 1. Introduction

ALS is a fatal disease caused by progressive degeneration and death of skeletal motor neurons (MN). Typical pathological hallmarks of ALS are the impairment of lower and upper MNs leading to muscular weakness, limb spasticity, weight loss, and slow and distorted speech caused by tongue wasting [1]. As the disease progresses, ALS patients lose control of body movement and later on the ability to speak and breathe. Ninety percent of ALS patients die between three and five years after the appearance of the first symptoms as a consequence of respiratory failure [1]. More than 90% of all ALS cases are defined as sporadic ALS (sALS), and 5–10% are classified as familiar ALS (fALS). In the latter, mutations in several genes were directly linked to ALS disease (https://alsod.ac.uk/, accessed on 22 February 2022). Currently, the most abundant gene products associated with ALS include the transactive response DNA binding protein 43 kDa (TDP-43), fused in sarcoma (FUS), chromosome 9 open reading frame 72 (C9orf72), and superoxide dismutase 1 (SOD1) [2,3]. The discovery of a glycine to alanine mutation in the SOD1 gene in 1993 was the earliest description of a genetic cause associated with fALS [4]. Since then, more than 160 mutations in the SOD1 gene have been primarily associated with ALS (https://alsod.ac.uk/, accessed on 22 February 2022).

Today, different mutant SOD1 mouse models extraordinarily reflect the human ALS phenotype and can therefore be used as suitable experimental models to investigate the underlying molecular mechanisms for ALS, including cellular and molecular studies at pre-symptomatic stages [5]. The SOD1^G93A^ mouse model is the most commonly used mouse colony in ALS research, as it was discovered more than 25 years ago and is well established in ALS research [5]. SOD1^G93A^ transgenic mice express ubiquitously mutant human SOD1, replacing glycine for alanine in residue 93. Due to this, mice expressing mutant SOD1 develop a pathogenic phenotype that is comparable to fALS in humans [5].

Even if the exact molecular pathway leading to neurodegeneration is still unknown, there is a high likelihood that an interplay between a variety of pathologic cellular mechanisms contributes to the motor neuron loss observed in ALS patients [6]. Nowadays, it is known that the accumulation of mutant protein aggregates, abnormal RNA processing due to ER stress, mitochondrial dysfunction, oxidative stress, and the accumulation of reactive oxygen species (ROS) are malfunctions contributing to neurodegeneration during ALS progression [6].

In 2015, it was suggested that ALS disease is initiated by intrinsic injury of motor neurons, whereas disease propagation is driven by the activation of microglia and T-lymphocytes [7,8]. In the early stages of ALS, M2 microglia are present, and T2-helper (TH2) cells and regulatory T-cells (Tregs) take a protective role by sequestering anti-inflammatory molecules [8]. Therefore, M2 microglia promote tissue repair and support neuron survival [9]. On the contrary, advanced stages of ALS are characterized by M1 microglia, which release pro-inflammatory cytokines contributing to neuronal cell death, and cytotoxic T-cells (Tcyt) triggering neuroinflammation [8,9].

Despite the growing number of research projects aimed at deciphering the molecular pathomechanism of ALS, to date no single therapeutic approach has been found to significantly delay the progression of ALS or halt neurodegenerative processes. Therefore, one compelling approach is to target the endogenous antioxidant system of the body in an attempt to reduce inflammatory processes [10]. One effective molecule for modulating oxidative processes in vivo is ozone (O_3_). Due to its high reactivity, the oxidative and primarily toxic effect is narrowly limited to O_3_-tissue contact sites. This leads to dose-dependent lipid peroxidation, protein oxidation, and the generation of ROS, eventually leading to DNA damage and apoptosis [11]. Inhalation of ozone is known to cause serious damage to the lungs [12], and contact with cells after insufflation of an O_3_/O_2_ gas mixture into the peritoneal cavity will also result in direct toxic effects. Despite these negative effects, insufflation of an O_3_/O_2_ gas mixture into the peritoneal cavity was found to promote oxidative preconditioning and, thus, may reverse chronic oxidative stress by enhanced endogenous production of antioxidants, improvement of local blood circulation, and an induction/modulation of immune responses, which have been shown to be beneficial when used as a therapeutic strategy [13].

The beneficial effects of the insufflated O_3_/O_2_ gas mixture also modulate degenerative processes in the central nervous system. For cortical and cerebral neurons, O_3_/O_2_ therapy was able to attenuate age-related changes by reducing oxidative stress, apoptosis, and gliosis, as well as improving neurogenesis [14,15]. In APP/PS1 transgenic mice, an animal model for Alzheimer’s disease, repetitive O_3_/O_2_ treatment ameliorated behavioral and pathological deterioration and reduced the level of amyloid-β precursor protein in the prefrontal cortex and hippocampal formation [16]. Furthermore, O_3_/O_2_ therapy has already been applied in clinical studies examining its effects on patients suffering from multiple sclerosis. These trials strikingly demonstrated the antioxidant characteristics and anti-inflammatory and immunomodulatory effects of O_3_/O_2_ therapy by reducing oxidative stress in the context of brain disease [17,18].

Along this line, we investigated the therapeutic potential of an insufflated O_3_/O_2_ gas mixture on disease progression and outcome, neurodegeneration, and innate and peripheral inflammation in the SOD1^G93A^ mouse model of ALS. We found that O_3_/O_2_ treatment at the beginning of the symptomatic disease phase leads to a select attenuation of neurodegeneration and innate neuroinflammation and preservation of lymph node monocyte counts, suggesting that medical ozone may positively modulate several disease parameters of this incurable disease.

## 2. Results

### 2.1. Therapeutic Treatment of SOD1^G93A^ Mice with Medical Ozone Does Not Affect Disease Progression, Motor Abilities, and Survival

To closely mimic the human situation, we aimed to therapeutically treat SOD1^G93A^ mice with medical ozone individually at the time when clinical symptoms became evident. Monitoring of body weight revealed that young SOD1^G93A^ and WT mice steadily gained weight over time, as expected (Figure 1A). The weights of SOD1^G93A^ mice, however, plateaued at around P77, and did not drop significantly until P126. Even until the end-stage, ozone- and sham-treated SOD1^G93A^ mice showed no significant differences in body weight between each other, nor to their respective WT controls. Hence, body weight was not a useful indicator of disease onset and progression in individual mice, and thus the appearance of clinical symptoms was rated by a combination of two tests that assessed muscle performance [19]. While a TST score >0 was observed at P80 in the sham-treated (range: P63–P133) and at P84 in the ozone-treated (range: P63–P98) SOD1^G93A^ animals (Figure 1B), a PaGE score > 0 was observable at day P126 in the sham (range: P85–P140)—and at P119 in the O_3_/O_2_ (range: P84–P147)—treated SOD1^G93A^ mice (Figure 1C). Thus, not to limit the time window between the start of treatment and the expected death of the animals too much, our therapeutic treatment regime was initiated the latest at P110 even if a PaGE score was still not evident at that time. In summary, the ages of our study mice at the beginning of the 5-day medical ozone treatment regime ranged from P85–P110. The symptomatic disease phase was characterized by a similar progressive loss of motor abilities, both in sham- and ozone-treated mice. The sham-treated mice reached a final PaGE score of 5 with a median of 136 days (range: 120–151), while the ozone-treated mice reached this stage with a median of 139 days (range: 124–164, *p* = 0.26, Log-rank test). In addition, the mean disease duration, defined as days from the first day of treatment to death, was the same in sham- (40.42 ± 2.3 days, mean ± SEM) and medical ozone-treated (40.33 ± 2.1 days, mean ± SEM) animals (*p* = 0.98, Figure 1E). Finally, a comparison of the median survival of sham-treated SOD1^G93A^ mice (median = 146 days, range: 126–164) with that of medical ozone-treated animals (median = 150 days, range: 133–166) did not differ (*p* = 0.54, Figure 1D).

### 2.2. Therapeutic Treatment of SOD1^G93A^ Mice with Medical Ozone Decelerates Motor Neuron Degeneration in the Trigeminal Motor Nucleus

To evaluate the extent of neuronal degeneration in the core motor nuclei of the brainstem (Figure 2A), the number of ChAT-immunoreactive (ir) neurons was counted in these areas and compared between the experimental groups. Comparisons were made between WT and SOD1^G93A^ mice at P60, together with sham- and medical ozone-treated SOD1^G93A^ mice at P130. No differences in the numbers of ChAT-ir neurons between groups were observed for the oculomotor (III, Figure 2B; WT P60 = 24.65 ± 3.76, SOD1 P60 = 22.30 ± 3.64, SOD1 sham P130 = 18.44 ± 3.86, SOD1 ozone P130 = 28.16 ± 6.36), the abducens (VI, Figure 2D; WT P60 = 60.91 ± 4.67, SOD1 P60 = 57.95 ± 2.91, SOD1 sham P130 = 55.21 ± 0.16, SOD1 ozone P130 = 57.23 ± 4.64), and the dorsal nucleus of the vagus nerve (X, Figure 2F; WT P60 = 14.93 ± 2.44, SOD1 P60 = 12.76 ± 4.60, SOD1 sham P130 = 7.99 ± 1.53, SOD1 ozone P130 = 9.10 ± 0.62). In the hypoglossal nucleus (XII, Figure 2G; WT P60 = 27.23 ± 0.75, SOD1 P60 = 21.19 ± 3.31, SOD1 sham P130 = 15.99 ± 0.96, SOD1 ozone P130 = 15.13 ± 1.58), only trends toward decreasing numbers of ChAT-ir neurons in the two SOD1^G93A^ groups at the late stage of disease were visible. Robust effects of neuronal degeneration between P60 and P130 SOD1^G93A^ mice were seen in the trigeminal motor nucleus (V, Figure 2C; WT P60 = 11.54 ± 1.27, SOD1 P60 = 12.41 ± 2.17, SOD1 sham P130 = 3.90 ± 0.57, SOD1 ozone P130 = 7.90 ± 0.22), in the facial nucleus (VII, Figure 2E; WT P60 = 14.51 ± 1.26, SOD1 P60 = 15.84 ± 1.16, SOD1 sham P130 = 9.54 ± 0.75, SOD1 ozone P130 = 8.80 ± 0.37), and in the ambiguus nucleus (Figure 2H; WT P60 = 50.37 ± 0.91, SOD1 P60 = 46.98 ± 2.74, SOD1 sham P130 = 27.67 ± 2.83, SOD1 ozone P130 = 21.25 ± 4.23). Interestingly, an attenuating effect of medical ozone on motor neuron degeneration was observed in the trigeminal motor nucleus only (*p* = 0.0025).

### 2.3. Therapeutic Treatment of SOD1^G93A^ Mice with Medical Ozone Attenuates SOD1-Related Vacuolization in the Trigeminal Motor Nucleus

A characteristic feature of human SOD1^G93A^-related neuropathology in the mouse model used is the appearance of intraneuronal vacuoles, which are most likely caused by expansion of the intermembrane space of mitochondria [20,21,22]. Since these vacuoles are fluid-filled, they are potentially accessible in live animals using T2-weighted MRI analysis. Because of their sizes, the trigeminal (Figure 3A) and facial (Figure 3B) motor nuclei were readily detectable in SOD1^G93A^ mice at the end stage, when compared to the WT. The level of vacuolization of these ALS-vulnerable nuclei was quantified with anti-human SOD1 antibodies in immunohistochemical analysis [23]. As expected, human SOD1-related vacuolization was already observed at P60 (V: 4.60 ± 1.21% area, VII: 5.50 ± 0.75%), and a strong increase in area covered by human SOD1 immunoreactivity was evident for both the trigeminal (Figure 4B, 19.70 ± 0.82, *p* = 0.0002 when compared to P60) and facial (Figure 4C, 25.50 ± 2.07, *p* = 0.0026 when compared to P60) nuclei in sham-treated SOD1^G93A^ mice at the disease end-stage. Human SOD1-related vacuolization was attenuated after medical ozone treatment in the motor trigeminal nucleus at end-stage (15.30 ± 2.81) when compared to sham treatment (*p* = 0.03) but not in the facial motor nucleus (25.00 ± 1.00, *p* = 0.42).

### 2.4. Therapeutic Treatment of SOD1^G93A^ Mice with Medical Ozone Attenuates Microglial Reactions

To assess the extent of brain innate inflammation, immunohistochemical staining for Iba1 was performed to detect reactive microgliosis, and for GFAP to detect astrogliosis. Quantification of microglia cells in the brain stem revealed no difference between the WT and SOD1^G93A^ animals at P60 (Figure 4A, 0.67 ± 0.25 vs. 0.35 ± 0.1, *p* = 0.16). A significant increase in the number of Iba1-ir cells was observed in sham-treated SOD1^G93A^ animals at P130 (Figure 4B, 3.82 ± 0.80) when compared with P60 SOD1^G93A^ animals (*p* = 0.0009). In contrast, animals treated with medical ozone (1.21 ± 0.64) showed a much smaller increase in Iba1-ir cells than sham-treated animals at P130 (*p* = 0.02). Even at this advanced disease stage, microgliosis only slightly increased in animals treated with medical ozone compared to SOD1^G93A^ animals at P60 (*p* = 0.31). To determine the extent of astroglia hypertrophy by GFAP immunohistochemistry, the proportion of the GFAP-ir area was determined in relation to the total area of each brain region analyzed (Figure 4D). While the GFAP-ir area was found not to increase at P60 in SOD1^G93A^ animals when compared to the WT (3.01 ± 2.48 vs. 2.28 ± 1.07, *p* = 0.68), pronounced astrogliosis was measured in the sham-treated SOD1^G93A^ animals in the advanced disease stage at P130 (Figure 4E, 26.21 ± 8.79, *p* = 0.0047). In contrast, animals treated with medical ozone displayed an attenuated astrogliosis at P130 (Figure 4F, 16.75 ± 5.61); however, this was neither significantly reduced when compared with the same-aged sham-treated SOD1^G93A^ animals (*p* = 0.14) nor significantly increased when compared with the non-treated SOD1^G93A^ animals at P60 (*p* = 0.16).

### 2.5. Therapeutic Treatment of SOD1^G93A^ Mice with Medical Ozone Preserves Monocyte Counts in Mesenteric Lymph Nodes

In addition to motor function tests, the leukocyte cell composition was examined by FACS in whole blood, spleen, and mesenteric lymph nodes in end stage SOD1^G93A^ mice to reveal possible effects of medical ozone treatment on the immune status (Figure 5). To determine the normal leukocyte composition in young adult wild-type and SOD1^G93A^ mice, blood leukocytes and leukocytes derived from homogenates of the spleen and mesenteric lymph nodes were examined on day P60. At this time, there was already a significantly reduced number of monocytes in the SOD1 animals in all samples examined (Figure 5). The percentage of lymphocytes and granulocytes remained unchanged.

At the end stage of the disease, relative numbers of lymphocytes, granulocytes, and monocytes in total blood (Figure 5A) as well as spleen (Figure 5B) of SOD1^G93A^ mice showed no change between sham- or O_3_/O_2_-treated animals. In addition, no change in the composition of the leukocyte subpopulations could be measured compared with young SOD1^G93A^ animals at P60. In contrast, a significant reduction in the number of monocytes was found in the mesenteric lymph nodes of sham-treated SOD1^G93A^ animals at the terminal disease stage, both compared with O_3_/O_2_-treated SOD1^G93A^ animals at the end-stage and in young untreated SOD1^G93A^ animals at day P60 (Figure 5C). Again, the number of lymphocytes and granulocytes in the mesenteric lymph nodes did not change in an age- or treatment-dependent manner.

## 3. Discussion

Ozone is considered a toxic agent with irritating effects, especially in the respiratory tract [24]. Recent findings, however, indicate that small and precisely calculated amounts of medical ozone, i.e., ozonized oxygen, can have positive effects in the body by increasing systemic antioxidant status [25]. Several studies have revealed that low doses of medical ozone may be effective in the treatment of a variety of chronic conditions related to inflammation and oxidative stress [26]. In 2008, Re et al. [27] reported that medical ozone has therapeutic neuroprotective potential when used in an in vivo model of Parkinson’s disease. Subsequently, the same group pointed out that the health benefits of medical ozone therapy may be due to an increase of systemic antioxidants, reduction of oxidative damage, improvement of oxygen blood transportation and delivery, as well as anti-inflammatory effects [28]. Thus, because of its anti-inflammatory and antioxidant-inducing capacity, medical ozone therapy could also be an interesting disease-modifying intervention in ALS.

In our experimental setup, we treated SOD1^G93A^ mice at their individual symptom onset. Our primary goal was considered a preclinical drug test that should potentially translate positive results in animals into the human situation [29] and thus favored a symptomatic over a pre-symptomatic treatment start [30]. A dose of 50µg ozone per ml and a five-day *i.p*. treatment regime were chosen because this combination had already been shown to be effective in our own cancer and sepsis studies [31,32]. Unfortunately, we did not observe a beneficial effect on disease duration, decline of body weight and motor functions, or survival. On the other hand, medical ozone did not worsen disease in this animal model and did not produce unfavorable side effects, i.e., painful pneumoperitoneum [33]. Given the disease-modulating effects described below, experimental adjustments with respect to e.g., treatment duration, route of administration, and ozone concentration should be considered in future trials.

Monitoring neuronal degeneration in ALS-vulnerable and -resistant brainstem motor nuclei revealed a select slowing down of trigeminal motor neuron loss in response to treatment with medical ozone. Other ALS-vulnerable nuclei, i.e., the facial, hypoglossal, and ambiguous nuclei, showed the expected variable levels of degeneration [34,35] that, however, were not susceptible to treatment. While the reason for this unresponsiveness remains to be determined, it may indicate differences in intrinsic responsiveness to treatment. Deciphering the nature of this phenomenon may unravel strategies that help protect all ALS-vulnerable motor neurons from extended degeneration, and ultimately preserve motor abilities for a longer time. Additional analysis of ozone effects on upper motor neuron pathology in the M1 cortical area is equally warranted because this compartment is also prone to degenerate, both in humans [36] and in animal models [37]. A select preservation of those neurons that are needed for mastication indicates a beneficial effect of medical ozone on food intake and, subsequently, energy homeostasis.

When analyzing mutant SOD1-related vacuolization in two ALS-vulnerable brainstem motor nuclei, we observed an attenuation of pathologic vacuoles in the trigeminal motor nucleus after treatment with medical ozone when compared to sham-treated SOD1^G93A^ mice. These vacuoles are most likely swollen mitochondria, with the involvement of aggregated mutant SOD1 for their appearance [20,21,38]. Since pathologic aggregation of proteins is considered a main reason for neurodegeneration in general [39], and SOD1-related neurodegeneration in ALS in particular [40], reductions in the severity of this process may be the underlying reason for our observed protection against trigeminal motor neuron loss. We also showed that MRI analysis of the trigeminal and facial motor nuclei recapitulated human SOD1-related vacuolization. Thus, a non-invasive neuroimaging of ALS-vulnerable brain nuclei proved suitable as a possible surrogate measure in further studies [41].

Besides selecting preservation of motor neuron counts, we also discerned an effect of medical ozone on innate inflammation. Microglia expansion was attenuated after medical ozone treatment, while astrocyte hypertrophy remained largely unaffected. Microglia are the resident immune cells of the brain, with key functions in immune surveillance and host defence [42]. Strong and chronic expansion of the pool of microglia seen in sham-treated mice may be associated with a switch from a neuroprotective to a neurodestructive functional state [43]. While molecular proof warrants further investigation, we here propose that microglia, after medical ozone treatment, may still elicit beneficial effects on motor neuron survival.

During the last two decades, it has become increasingly clear that the systemic immune response may essentially participate in mitigating brain damage in neurodegenerative disease [44]. For example, immune cell-deficient SOD1^G93A^ mice progressed faster to the end stage compared to their immune-competent littermates [45,46], and a passive transfer of T cells (T_eff_ and T_reg_) to SOD1^G93A^ mice was able to delay loss of motor function and extended survival [47]. Furthermore, reductions as well as elevations in a range of immune cell subpopulations have been shown for blood from ALS patients (recently reviewed by [48]). In our study, we found that, on a gross cellular level, lymphocyte and granulocyte populations did not show disease- or treatment-related changes or adaptations. The numbers of monocytes, however, were found reduced already in SOD1^G93A^ mice at P60 in blood, spleen, and mesenteric lymph nodes, when compared to WT. Medical ozone treatment prevented a further reduction of the monocyte population in the mesenteric lymph nodes, but not in the blood or spleen. While a contribution for e.g., subtypes of lymphocytes or granulocytes in ALS neurodegeneration after medical ozone treatment warrants further investigation, the observed preservation of monocyte numbers suggests that this immune cell population was readily accessible to treatment, and this effect may be important in ALS. SOD1^G93A^ mice that have undergone bone marrow transplantation from mice deficient in myeloid differentiation primary response protein 88, MyD88, exhibited an earlier disease onset and shorter survival compared to mice that received a normal bone marrow transplant [49]. In addition, a positive correlation was observed between blood non-inflammatory monocytes and survival in the SOD1^G93A^ mouse model [50].

From a mechanistic point of view, the observed effects of medical ozone on neuronal survival and on the brain’s innate immune system in our study mice most likely are not direct, but rather indirect. Once applied to the peritoneal cavity, ozone reacts locally with biomolecules and then quickly disappears. An uptake into the circulation with subsequent transport to the CNS is highly unlikely. Hence, the therapeutic efficacy of ozone may be due to the production of local and moderate oxidative stress, serving as a preconditioning effect on endogenous pro-antioxidant mechanisms [51]. Either ozone acts as a modulator by inducing secondary messengers, e.g., reactive oxygen species (ROS) and lipid oxidation products (LOPs), that may reach the CNS via the bloodstream and elicit effects on neurons and glia. Or it `activates´ resident monocytes in mesenteric lymph nodes, which in turn infiltrate the brain parenchyma i.e., in the area of the spinal cord and brain stem, differentiate locally into macrophages, and terminate detrimental microglia reactions [44]. Thus, preventing a decline in monocyte numbers and activity via local application of medical ozone should be taken into consideration. As the routes of ozone application are diverse, e.g., insufflation into the peritoneum or rectum, or *i.v*. administration into the blood in the context of autohemotherapy, the most efficient therapeutic strategy still has to be determined.

## 4. Materials and Methods

### 4.1. Compliance with Ethical Standards

All animal procedures were conducted in accordance with international standards on animal welfare, the European directive 2010/63/EU, and the German Animal Protection Law under a protocol approved by the county administrative government authorities in Giessen, Germany (V54-19c205h01 IVR20/26 no. G5/2018).

### 4.2. Mouse Strain, Handling, and Genotyping

Transgenic mice of the congenic strain B6.Cg-Tg(SOD1*G93A)1Gur/J (stock no. 004435, The Jackson Laboratory, Bar Harbor, ME) carrying human SOD1 with the pathogenic G93A mutation (SOD1^G93A^) in a high copy number were used [5]. Progeny for experimental analyses were obtained from five breeding pairs between SOD1^G93A^ males and C57BL/6J wild-type (WT) females. All mice, separated by gender, were housed in groups of two to six animals per cage on a 12 h dark/light cycle with unrestricted access to food and water. Mice were weighed before noon on a weekly basis. From the time when the SOD1^G93A^ mice showed motor deficits, additional moistened food and water were placed on the cage floor. SOD1^G93A^ animals were euthanized by an overdose of inhaled isoflurane (Baxter, Unterschleissheim, Germany) when at least one of the following termination criteria were reached: (*i*) weight loss of >20% from maximum, or (*ii*) holding time in the Paw Grip Endurance (PaGE) test < 20sec (see below). In addition, their general condition, appearance, and respiration were monitored at least weekly to register non-ALS-related issues that eventually led to an exclusion from the study.

Together with weaning at the age of 19–21 days postnatally (P), ear-punches were collected in 100 µL DirectPCR Lysis reagent (peqlab, Erlangen, Germany), supplemented with 2 µL of 20 mg/mL proteinase K (Sigma-Aldrich, Steinheim, Germany), and incubated for 12–18 h at 56 °C. After this lysis, the samples were incubated for an additional 45 min. at 85 °C, and cell debris pelleted for 1 min at 13,000 rpm. The subsequent PCR was performed essentially according to the protocol supplied by The Jackson Lab for this mouse strain, using KAPA 2G Fast Ready Mix (peqlab). All transgenic mice identified by this procedure were further controlled for transgene copy number by semi-quantitative PCR using fluorescent probes. In detail, each PCR reaction contained 10 µL of PCR master mix (KAPA PROBE FAST qPCR Master Mix (2X) ABI Prism, Kapa Biosystems, Wilmington, MA, USA), 2 µL primer mix (10 µM), 4 µL H_2_O, and 4 µL genomic DNA (1:10 dilution taken from the ear punch sample). The primer mix contained forward and reverse primers for the detection of the human SOD1 transgene (TG) and an internal positive control (IPC), and probes for TG (FAM as reporter, and BHQ1 as quencher) and IPC (JOE as reporter, and BHQ1 as quencher) detection. Primer and probe sequences were essentially as recommended by The Jackson Laboratory. The PCR reactions were performed in triplicate for each sample using 96-well plates (Biozym Scientific, Hessisch Oldendorf, Germany), a protocol supplied by The Jackson Laboratory, and run in an ABI PRISM 7900HT device (Applied Biosystems, Darmstadt, Germany). For all PCR reactions, a defined cycle threshold (ct) was set, the mean from the triplicate reactions was calculated, and a delta-ct (ct TG-ct IPC) was calculated. All values obtained from the transgenic offspring were compared to the mean delta-ct of the five SOD1^G93A^ breeder male mice. A difference >0.5 delta-ct had to be considered questionable because of a possible transgene copy loss during meiosis. If these mice turned out to live longer than two standard deviations from the mean for that particular group in the survival analysis, they were excluded retroactive [52]. However, none of our experimental mice fell into this category.

### 4.3. Experimental Groups and Ozone Treatment Paradigms

Both SOD1^G93A^ transgenic and C57BL/6J WT littermates were randomly divided into the following experimental groups: (i) ozone-treated animals that received an O_3_/O_2_ gas mixture therapy (SOD1:ozone, WT:ozone); (ii) sham-treated animals that received a puncture into the peritoneum only (SOD1:sham, WT:sham); (iii) untreated animals (SOD1:non-treated, WT:non-treated. Overall, 42 SOD1^G93A^ transgenic and 43 C57BL/6J WT control mice were used in the study. All experimental groups included an equal number of male and female mice. For the generation of the O_3_/O_2_ gas mixture, an ozone gas processor (MedozonIP, Herrmann Apparatebau, Kleinwallstadt, Germany) was used. Animals of the ozone-treatment groups received insufflations of 1 mL O_3_/O_2_ gas mixture containing 50 μg O_3_ per milliliter (1 mg/kg body weight), corresponding to a contingent of 2.5% O_3_ and 97.5% O_2_, into the peritoneum. The ozone-treatment paradigm consisted of i.p. injections of 1 mL O_3_/O_2_ gas mixture for five consecutive days [31].

### 4.4. Behavioral Tests

Two tests evaluating motor function/dysfunction were performed according to the recommendations of Leitner and colleagues [52]. In the tail suspension test (TST), mice were picked up by their tails and held steadily for several seconds. Evenly spread hind legs showed an intact motor system (score = 0), whereas motor deficits appeared due to a non-symmetrical spreading of the legs and a clearly perceptible tremor in at least one leg (score = 1). The paw grip endurance (PaGE) test is a common assay used to examine the motor strength and endurance of the mouse hind limbs [53]. In this test, mice were placed on a rotating metal grid and turned upside down after finding a foothold. Mice without motor deficits showed no difficulty in maintaining their own body weight on the grid for a period of 120 sec. (score = 0). SOD^G93A^ mice suffering from disease-related muscle wasting are not able to hold onto the grid, and with progression of the disease, the holding time decreased (<120 s> 100 s, score = 1; <20 s, score = 5). The PaGE test was repeated three times, giving the mice time to rest after each trial, and the best result was scored. In all behavioral testing, the observer was blinded to the genotype and treatment groups.

### 4.5. Magnetic Resonance Imaging

A 7 Tesla magnetic resonance (MRI) scanner for small animals (Clinscan 70/30 URR, Bruker) was used. Mice were euthanized by a lethal dose of inhaled isoflurane and positioned in the isocenter of the MRI scanner. MRI acquisition settings were as follows: Repetition time: 2500 ms; Echo time: 29 ms; Field of view: 25 mm; Matrix: 256 × 256; Layer thickness: 0.1 mm; Flip angle: 108°; and Voxel size: 0.1 × 0.1 × 0.1. MRI images were analyzed using SYNGO FY DICOM Viewer software. Horizontal and coronar T2-weighed spin echo images were obtained for analysis.

### 4.6. FACS Analysis

EDTA-blood was collected by cardiac puncture immediately after mouse euthanasia, and the spleen and the complete small intestine mesentery were removed. Mesenteric lymph nodes were then isolated under a binocular (Olympus SZH10). Spleen and lymph nodes were then crushed in 500 µL Ca^++^/Mg^++^-free PBS using an electronic microtube homogenizer (KIMBLE^®^ PELLET PESTLE^®^ Cordless Motor; DWK Life Sciences, Mainz, Germany), and the homogenates were passed through a 40 µm sieve (pluriSelect Life Science, Leipzig, Germany). Erythrocytes in the spleen homogenate and EDTA blood sample were largely removed by hypotonic lysis. For this, 400 µL blood or 500 µL spleen homogenate were incubated for 10 min at room temperature in 15 mL lysis buffer containing 155 mM ammonium chloride, 10 mM potassium bicarbonate, and 1 mM EDTA. For the lymph nodes, the hypotonic lysis step was omitted. After centrifugation of the hypotonic lysis mixtures (400 rcf, 10 min at 4 °C), the cell pellets and the sieved lymph node homogenate were washed with 12 mL Ca^++^/Mg^++^-free PBS, centrifuged again, and the pellets were resuspended in 1 mL each of Ca^++^/Mg^++^-free PBS containing 0.5% (*w/v*) paraformaldehyde. Finally, they were washed in 12 mL PBS. FACS analysis was performed on a BD LSR II Workstation (BD Biosciences, San Jose, CA, USA), and for evaluation, the FlowJo (7.6.5, Becton, Dickinson and Company, San Jose, CA, USA) program was used.

### 4.7. Immunohistochemisty

For histologic analysis, brain tissues were obtained from the same mice used in the MRI study. The mice were euthanized by isoflurane inhalation, and their brains were quickly dissected and fixated for 24 h at room temperature using Bouin–Hollande containing 4% (*w/v*) picric acid, 2.5% (*w/v*) cupric acetate, 3.7% (*v/v*) formaldehyde, and 1% (*v/v*) glacial acetic acid. Excess fixative was removed by repeated washing steps in 70% isopropanol for at least two weeks, followed by dehydration of the samples through an alcohol series of increasing concentrations and xylene. The samples were then embedded in paraffin, and 7 µm thick serial tissue slices were generated with a microtome. Histological stains on select sections were done with Giemsa staining solution. Single immunohistochemistry was performed as described previously [54]. Briefly, tissue sections were deparaffinized in xylene and rehydrated through a graded series of isopropanol, including 30 min of incubation in methanol/0.3% hydrogen peroxide to block endogenous peroxidase activity. Antigen retrieval was achieved by incubation in 10 mM sodium citrate buffer (pH 6.0) at 92–95 °C for 15 min. Nonspecific binding sites were blocked with 5% bovine serum albumin (BSA) in 50 mM phosphate buffered saline (PBS, pH 7.45) for 30 min, followed by an avidin-biotin blocking step (Avidin–Biotin Blocking Kit, Boehringer, Ingelheim, Germany) for 20 min each. Subsequently, primary antibodies were applied in PBS/1% BSA overnight at 16 °C, followed by 2 h at 37 °C. Primary antibodies included: goat-anti-ChAT (1:200 final dilution, order no. AB144P, Millipore, Burlington, MA, USA), guinea pig-anti-GFAP (1:10.000, Z334, Dako), rabbit-anti-Iba1 (1:1.000, 019-19741, FUJIFILM Wako Chemicals, Neuss, Germany), and goat-anti-human SOD1 (1:1.000, sc-8637, Santa Cruz Biotech.). After several washes in PBS, the sections were incubated for 45 min at 37 °C with species-specific biotinylated secondary antibodies (donkey-anti-goat or -guinea pig or -rabbit, Dianova, Hamburg, Germany), diluted 1:200 in PBS/1% BSA, washed, and incubated for 30 min with avidin-biotin-peroxidase complex (Vectastain Elite ABC kit; Vector Laboratories, Burlingame, CA, USA). Immunoreactions were then visualized by an 8 min incubation in 3,3-diaminobenzidine (DAB, Sigma Aldrich, Deisenhofen, Germany), enhanced by the addition of 0.08% ammonium nickel sulfate (Fluka, Bucks, Switzerland). After washes in distilled water, the sections were dehydrated through a graded series of isopropanol, cleared in xylene, and mounted under coverslips. Digital bright-field pictures were taken with a Zeiss Axio Imager M2 microscope (Zeiss, Oberkochem, Germany) equipped with a Zeiss AxioCam HRC camera and ZEN Lite Analyses software (Zeiss).

### 4.8. Quantitative Procedures and Statistical Evaluations

Individual mouse body weights were normalized to 100% at day P35 to compensate for differences in absolute body weight between male and female mice and between the experimental groups analyzed. Statistical differences in body weight were calculated using the regular two-way ANOVA with Tukey correction. The TST, PaGE, and survival scores were analyzed with Log-Rank tests and plotted in Kaplan–Meier curves. The disease time duration was determined as the time from the first day of treatment to euthanization at the end stage and subjected to an unpaired t-test analysis. An unpaired two-tailed t-test was used to evaluate the FACS data. The evaluation of neurodegeneration in brainstem motor nuclei was done by counting ChAT-immunoreactive (ChAT^+^) nerve cell bodies. The activation status of astroglia and microglia of the brainstem as well as the grade of vacuolization were evaluated on immune-stained sections for GFAP, Iba1, and human SOD1, respectively. The areas covered by immunoreactivity were determined with ImageJ software (version 1.52a, National Institutes of Health, Bethesda, MD, USA). For each of these histological analyses at least four sections through each brainstem motor nucleus, separated by at least 30 µm to avoid double-counting, were evaluated, and values normalized to the area of the nucleus in the respective section. To test for differences in brainstem nuclei, a one-tailed, unpaired t-test was applied. If variances were statistically different between the groups, a one-tailed unpaired t-test with Welch’s correction was applied. For comparisons of different analysis time points within a group, a one-sided *p* value was calculated; when comparing the two mouse lines (SOD1 vs. WT) or the different treatment groups, a two-sided *p* value was calculated. All statistical analyses were performed using GraphPad Prism 6 software (GraphPad Software, Inc., San Diego, CA, USA).

## 5. Conclusions

Given that the currently available ALS drugs, riluzole and edaravone, only show modest beneficial effects in patients, additional treatment options are needed [55]. We believe that medical ozone, if properly formulated and the best route of application determined [11], could serve as a simple, low-cost adjunct therapy for ALS. Although ozone will never reach the CNS by itself when peripherally applied, its immune-modulating capacity is intriguing and warrants full evaluation at the cellular and molecular levels. Deciphering the specific effects that are elicited on different types of leukocytes at the sites of administration will provide useful information about the mechanism by which medical ozone indirectly affects motor neuron survival and glia reactions in the CNS. In addition, ALS mouse models distinct from the used mutant human SOD1^G93A^ line should also be evaluated in future experiments to provide information about a more generalized capacity of medical ozone, and before attempting to translate this treatment paradigm into humans.

## Figures and Tables

**Figure 1 ijms-23-03403-f001:**
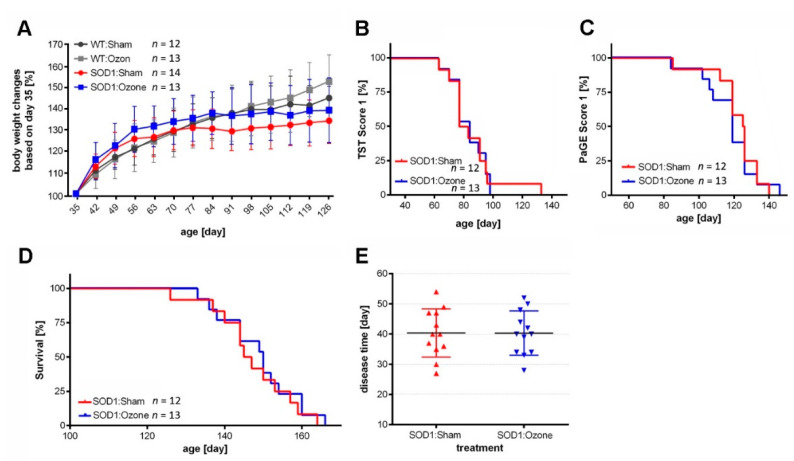
Clinical assessment of the study mice. (**A**) Changes in body weight normalized to 100% at P35. Values represent mean ± S.D. *p* = n.s. between groups, two-way ANOVA with Tukey correction. (**B**) Tail suspension test (TST) evaluating the beginning of motor deficits in the hind limbs (score = 1), *p* = 0.968, n.s., Log-Rank test. (**C**) Paw grip endurance test (PaGE) evaluating motor strength and endurance (score = 1), *p* = 0.672, n.s., Log-Rank test. (**D**) Survival analysis, *p* = 0.565, n.s., Log-Rank test. (**E**) Scatter plot (mean ± S.D.) of disease duration in days from first day of treatment to end-stage in sham- or ozone-treated SOD1^G93A^ mice. *p* = 0.979, n.s., unpaired *t*-test.

**Figure 2 ijms-23-03403-f002:**
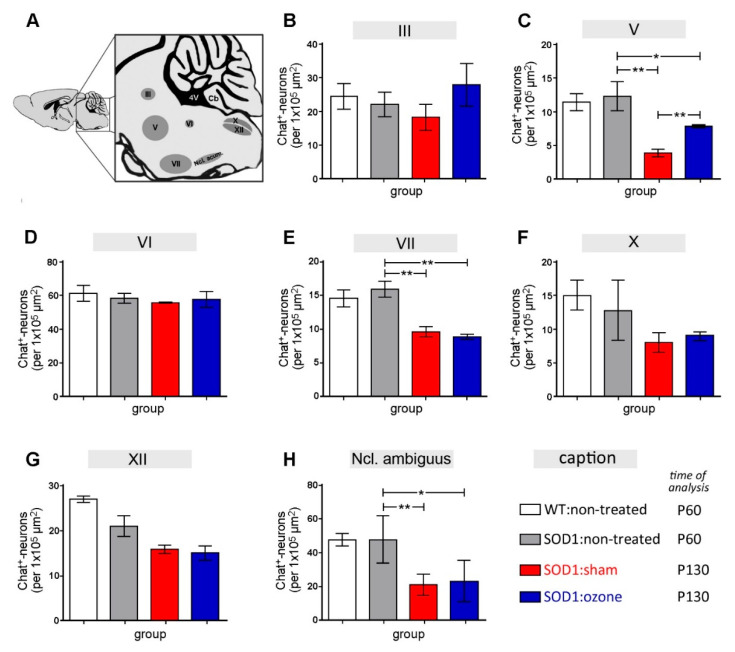
Effect of ozone treatment on motor neuron degeneration in the brainstem of SOD1^G93A^ mice. (**A**) Diagram showing the location of brainstem motor nuclei analyzed. (**B**–**H**) Evaluation of motor neuron degeneration in the (**B**) oculomotor (III), (**C**) trigeminal (V), (**D**) abducens (VI), (**E**) facial (VII), (**F**) vagal (X), (**G**) hypoglossal (XII), and (**H**) ambiguus nuclei by determining the number of ChAT-ir cell bodies within each nucleus. The identity of the four experimental groups is shown under the caption. Differences between groups (*n* = 3 for all groups) were analyzed using an unpaired *t*-test with Welch’s correction. Statistical differences (* *p* < 0.05; ** *p* < 0.01).

**Figure 3 ijms-23-03403-f003:**
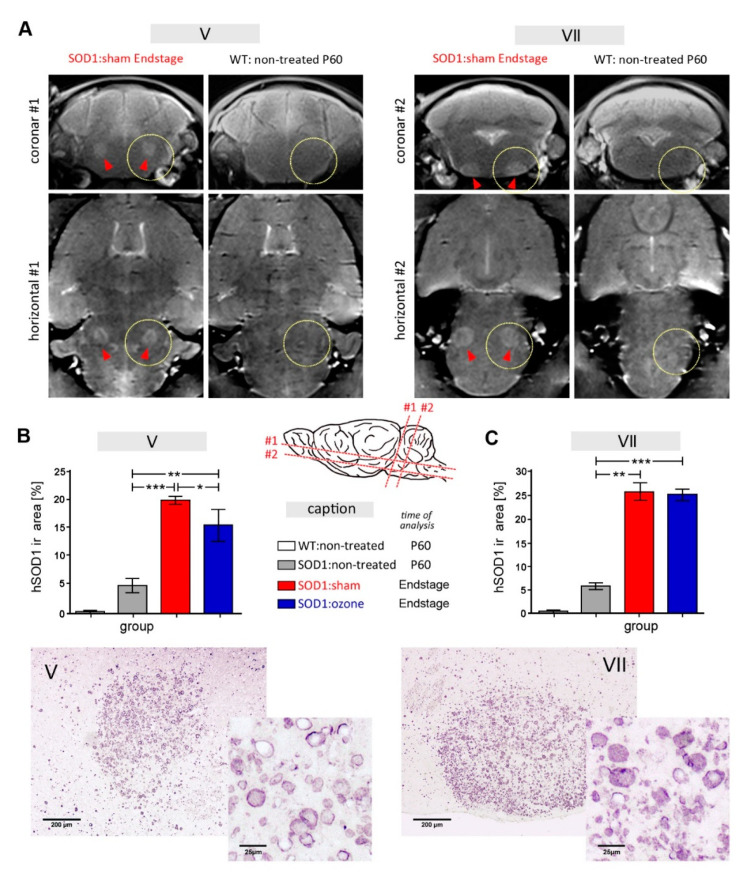
MRI and immunohistochemical analysis of human SOD1-related vacuolization in brainstem motor nuclei. (**A**) MRI analysis of SOD1^G93A^ mice revealed a marked increase in T2-weighed signals in the trigeminal (left four images, #1) and facial (right four images, #2) motor nuclei, shown in coronary and horizontal planes according to drawing. Arrowheads indicate the position of motor nuclei V and VII, respectively, in the yellow circled area from a representative MRI of a SOD1^G93A^ mouse at end stage. For comparison, MRIs of C57BL/6J wild-type (WT) mice are shown. (**B**,**C**) Quantification of human SOD1-immunoreactivity in the trigeminal (**B**) and facial (**C**) nuclei of WT and SOD1^G93A^ mice at an early preclinical age (P60) and at end stage. Representative immunohistochemical stains of brain nuclei in end-stage disease taken from sham-treated mice are shown below each graph. Data are presented as mean ± SEM (*n* = 3 for all groups). Differences between groups were analyzed using the Wilcoxon–Mann–Whitney test, with * *p* ≤ 0.05, ** *p* ≤ 0.01, *** *p* ≤ 0.001.

**Figure 4 ijms-23-03403-f004:**
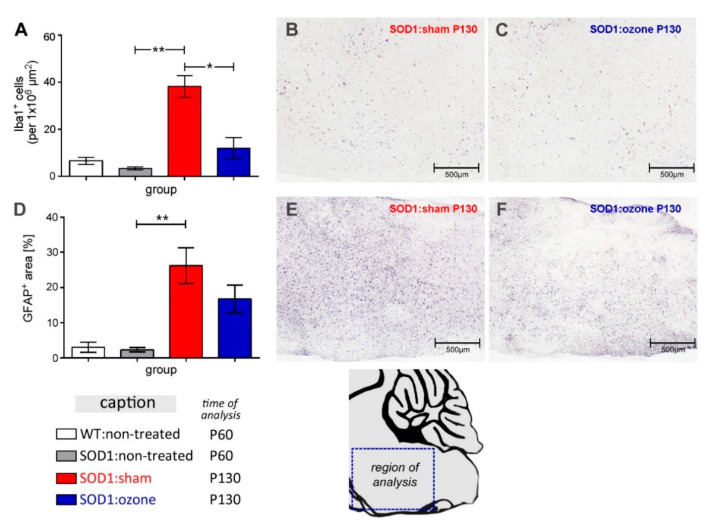
Neuroinflammation in the brainstem. Evaluation of (**A**–**C**) Iba-1 immunoreactivity as a measure for microglia proliferation, and (**D**–**F**) GFAP-ir for astroglia hypertrophy. Brainstem cartoon shows the region of quantitative analysis depicted in **A** and **D,** as well as the microscopic images of representative immunostaining for Iba1 (**B**,**C**) and GFAP (**E**,**F**). Differences between groups (*n* = 3 for all groups) were analyzed using an unpaired t-test with Welch’s correction. *, *p* < 0.05; **, *p* < 0.01.

**Figure 5 ijms-23-03403-f005:**
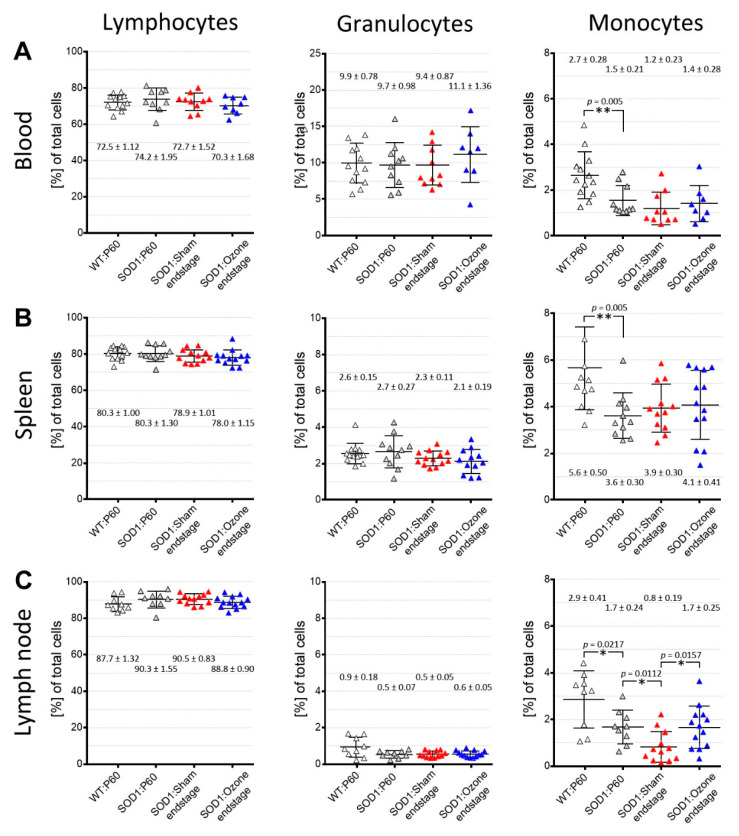
Leukocyte counts in blood, spleen, and mesenteric lymph nodes from WT and SOD1^G93A^ mice. Leukocytes were isolated from blood (**A**), spleen (**B**), and mesenteric lymph nodes (**C**), and the percentages of lymphocytes, granulocytes, and monocytes were determined by FACS analyses. Isolates derived from sham- and ozone-treated SOD1^G93A^ mice at the end stage were compared to isolates from wild-type (WT) and SOD1^G93A^ mice at P60. Scatterplots show the mean values ± standard deviations. An unpaired two-sided *t*-test was performed for the statistical calculation. Significance level: * *p* < 0.05; ** *p* < 0.01.

## Data Availability

Not applicable.

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
