# Peer review of "Therapeutic Treatment of Superoxide Dismutase 1 (G93A) Amyotrophic Lateral Sclerosis Model Mice with Medical Ozone Decelerates Trigeminal Motor Neuron Degeneration, Attenuates Microglial Proliferation, and Preserves Monocyte Levels in Mesenteric Lymph Nodes"

_ijms, 2022, doi:10.3390/ijms23063403_

Round 1
Reviewer 1 Report
I have read the article by Bette et al. with great interest. This is a very well delivered study and a very well written article. I have only some minor comments:
- On most of the outcomes ozone did not induce significant changes. I wonder if the ozone concentration was too low or too much. Please, discuss how was the concentration chosen and how would it possibly translate to humans.
- A particular animal ALS model, deficient in SOD was used. I wonder if another ALS model was used, would ozone induce any significant effect? Please, discuss.
- I miss a conclusion paragraph. Please, provide one.
Author Response
Comments and Suggestions for Authors
I have read the article by Bette et al. with great interest. This is a very well delivered study and a very well written article. I have only some minor comments:
- On most of the outcomes ozone did not induce significant changes. I wonder if the ozone concentration was too low or too much. Please, discuss how was the concentration chosen and how would it possibly translate to humans.
Answer: We used a dose of 50 µg ozone per ml gas mixture (O3/O2), which corresponds to a total amount of 1.0 mg/kg body weight of mice. In this concentration range, previous studies have shown immune-mediated preventive and therapeutic effects in a rat sepsis model in the and in a rabbit tumor model. Compared to the 1 mg O3/kg body weight in the SOD1 mouse model, the amounts of ozone were 2.5 mg/kg in the rat and 4 mg/kg body weight in the rabbit. Furthermore, in a study on the pain perception of O3/O2 gas mixture insufflated into the peritoneum, we recently showed that this amount does not lead to visceral pain. This was a major argument for the amount of O3 used in our study mice (see references no. 31-33 in the manuscript). Indeed, the question of the best ozone concentration or absolute amount of insufflated ozone is an intriguing one, that has not yet been settled. Our ozone processor can produce O3 in a range from 2-80 µg/ml, which would allow to analyze higher ozone concentrations up to 1.6 mg/kg body weight without increasing the volume of gas to be insufflated, but also much lower doses. Future experiments should evaluate dose-dependent outcomes to pin down the optimal ozone concentration. The overall goal has to be that quality of life is preserved and survival extended after ozone treatment.
We now include additional statements on this issue in the discussion section (line 338), in the Materials and Methods section (line 505), and in the new Conclusions paragraph (lines 627-642).
With respect to treatment of human patients, currently applied protocols include rectal insufflation, blood autohemotherapy, direct application to wounds, and intraarticular injection. For all these routes of application, the ozone concentration may vary (see recent review by Smith et al., reference no. 11 in the manuscript). Comparable to our experimentation in mice, pneumoperitoneum would be a reasonable route of application in humans. However, effective ozone concentrations for this currently are unknown. The transferability of data from animal experiments into the human situation thus is an open question and the subject of numerous discussions, particularly with regard to the widespread use of genetically modified mouse models. However, it is a fact that the use of ozone therapy for a variety of different diseases in humans is increasing and that the data collected on the effect of ozone on e.g. tumors, cells of the immune system or metabolic parameters largely correspond to those observed in animal experimental studies. This indicates a transferability of ozone-mediated effects between animal models and humans, but in our opinion is not yet sufficient to justify clinical interventions, but provides a basis for possible alternative treatment therapies.
- A particular animal ALS model, deficient in SOD was used. I wonder if another ALS model was used, would ozone induce any significant effect? Please, discuss.
Answer: The SOD1-G93A mouse model is not deficient in SOD1, but rather overexpresses a mutant human SOD1 variant which results in a gain-of-function (Gurney, 1994). Although genetic models may eventually recapitulate only a minor proportion of all ALS cases, the SOD1-G93A mouse model still is the most useful and most extensively studied model. Testing our ozone treatment paradigm in other genetic ALS models (e.g. Profilin1-, TDP43-mutations) should be considered in the future to be able to judge if the use of ozone is more generally useful. Since ozone most likely does not target the mutated protein directly, but rather modulates e.g. immune reactions, ozone treatment may also prove efficient in other ALS mouse models.
We now include a statement on this issue in the new Conclusion section, lines 638-642.
- I miss a conclusion paragraph. Please, provide one.
Answer: We now provide the requested conclusion paragraph after the Materials and Methods section in the revised manuscript
Reviewer 2 Report
This manuscript describes the therapeutic potential of an insufflated O3/O2 gas mixture on ALS progression and outcome, neurodegeneration, and innate as well as peripheral inflammation, in the SOD1G93A mouse model of ALS. It is clearly and concisely written. The results are interesting. However, threes are several issues for the authors to take into account.
Major Issues
- What would be the reason that the authors did not analyze the primary motor area in the frontal cortex in addition to the brain stem?
- An observation made by the authors is very interesting, because the oculomotor
and abducens Ncls. were not affected by the mutation having this ALS model. It is well known and clinically important for the patients to preserve ocular movement, which is the only method for communication with other people. It could be due to a compensatory increase in Mn-SOD upregulation. But whatever the mechanisms may be, it is intriguing. If this important observation has not done before, some statements should be incorporated in the text, strengthening suitability of this transgenic mice as an ALS model.
Minor Issues
- What do mSOD and hSOD stand for? Maybe mouse and human SOD.? (65 and else)
- vinternational should read international(435)
Author Response
Comments and Suggestions for Authors
This manuscript describes the therapeutic potential of an insufflated O3/O2 gas mixture on ALS progression and outcome, neurodegeneration, and innate as well as peripheral inflammation, in the SOD1G93A mouse model of ALS. It is clearly and concisely written. The results are interesting. However, there are several issues for the authors to take into account.
Major Issues
- What would be the reason that the authors did not analyze the primary motor area in the frontal cortex in addition to the brain stem?
Answer: Both upper motoneuron (UMN) and lower motoneuron (LMN) pathology is evident in human ALS and in mouse models. Hence, we fully agree with the reviewer that an analysis of the UMN compartment should gather relevant information with regards to the extent of beneficial effects of our ozone treatment paradigm. However, we refrained from analyzing the cortical primary motor area because we were not in possession of a unique molecular marker to unequivocally identify e.g. layer 5 UMN in our mice. Marques et al. (Brain Sci. 2021) recently very nicely showed that UMN degeneration precedes that of LMN degeneration. They utilized retrograde labeling with Fluorogold from the spinal cord to control for identity of UMN, a technique currently not available to us. In our planned future treatment regimens, we will certainly add an evaluation of UMN, after consultation of local specialists in this technique.
We added in the Discussion, lines 359-62, a note to this important issue that reads: “Additional analysis of ozone effects on upper motoneuron pathology in the M1 cortical area is equally warranted, because this compartment is also prone to degenerate, both in humans [36] and in animal models [37]”.
- An observation made by the authors is very interesting, because the oculomotor and abducens Ncls. were not affected by the mutation having this ALS model. It is well known and clinically important for the patients to preserve ocular movement, which is the only method for communication with other people. It could be due to a compensatory increase in Mn-SOD upregulation. But whatever the mechanisms may be, it is intriguing. If this important observation has not done before, some statements should be incorporated in the text, strengthening suitability of this transgenic mice as an ALS model.
Answer: Robust resistance of outer eye muscle neuronal nuclei in the brain stem is a long-known feature in human patients, and also in mouse models, and has already been extensively studied (for a review see: Nijssen et al., Acta Neuropathol 133:863-885, 2017). Hence, additional elaboration on this aspect was not warranted.
Minor Issues
- What do mSOD and hSOD stand for? Maybe mouse and human SOD.? (65 and else)
Answer: mSOD1 = mutant SOD1, hSOD1 = human SOD1. We agree with the reviewer that these terms were not consistently used and may confuse the reader.
Thus, in the manuscript we now wrote out `mutant´ and `human´ wherever applicable.
- vinternational should read international(435)
Answer: We checked this and find that in our paper copy this word is spelled out correctly.
Reviewer 3 Report
Review comments are attached

Author Response
Reviewer #3
The present manuscript discusses the use of medical ozone to treat SOD1G93A ALS disease. According to the presented data, model mice, treated with medical ozone, exhibit deceleration of trigeminal motoneuron degeneration, attenuation of microglial proliferation, and preservation of monocyte levels in mesenteric lymph nodes.
The work was carried out competently, yet there exist concerns over several points that are remarked below:
- In the introduction, the statement “The discovery of a glycin to alanin mutation in the SOD1 gene in 1993 was the earliest description of a genetic cause associated with fALS [4]” should be corrected to read “The discovery of a glycine to alanine mutation in the SOD1 gene in 1993 was the earliest description of a genetic cause associated with fALS [4]”.
Answer: corrected.
- Throughout the manuscript, the term “motoneurons” should be replaced by “motor neurons”.
Answer: corrected.
- In the introduction, the statement “Contrarily, advanced stages of ALS are characterized by M1 microglia, which release pro-inflammatory cytokines …” should be corrected to read “On the contrary, advanced stages of ALS are characterized by M1 microglia, which release pro-inflammatory cytokines …”.
Answer: corrected.
- In lines 232-236 of the same section, the description of the statement “Human SOD1-related vacuolization was attenuated after medical ozone treatment in the motor trigeminal nucleus at end-stage (15.30 ± 2.81) when compared to sham treatment (p = 0.03), but not in the facial motor ……” was obscured by the Figure. The statement should be visible for evaluation.
Answer: we moved the figure one line further down, so that the statement will be visible for inspection.
- In the legend of Figure 3, the various subfigures should have capital letters, designating the experiments, in line with the graphs themselves.
Answer: corrected in all 5 figure legends.
- In section 2.4, “2.4. Therapeutic treatment of SOD1G93A mice with medical ozone attenuates 252 microglial reactions”, the discussion of the results projects figures 3A-3F. Such figures do not exist. Do the authors mean figures 4A-4F? Attention to detail should be given for a journal of this caliber.
Answer: We apologize for this mistake in referring to a wrong figure. Now corrected.
- In the legend of Figure 4, the various subfigures should have capital letters, designating the experiments, in line with the graphs themselves.
Answer: corrected (see query #5).
- In section 2.5, the statement “Again, the number of lymphocytes as well as granulocytes in the mesenteric lymph nodes did not changed in an age- or treatment-dependent manner.” should be corrected to read “Again, the number of lymphocytes as well as granulocytes in the mesenteric lymph nodes did not change in an age- or treatment-dependent manner.”.
Answer: corrected.
- In the legend of Figure 5, the various subfigures should have capital letters, designating the experiments, in line with the graphs themselves.
Answer: corrected (see query #5).
- In the discussion section, the statement “SOD1G93A mice that received bone marrow transplantation from mice deficient in myeloid differentiation primary response protein 88, MyD88, exhibited an earlier disease onset and a sorter survival compared to mice that received a normal bone marrow transplant [51].” should be re-written to read “SOD1G93A mice that have undergone bone marrow transplantation from mice deficient in myeloid differentiation primary response protein 88, MyD88, exhibited an earlier disease onset and a shorter survival compared to mice that received a normal bone marrow transplant [51].”.
Answer: corrected.
- The title of section 4.5 should be corrected to read “Magnetic Resonance Imaging”.
Answer: corrected.
Round 2
Reviewer 3 Report
The requested changes were made as per instrucitons provided